# Analysis of low resource setting referral pathways to improve coordination and evidence-based services for maternal and child health in Ethiopia

**Geletaw Sahle Tegenaw**[ID][1,2]*, **Demisew Amenu**[3], **Girum Ketema**[2], **Frank Verbeke**[1], **Jan Cornelis**[1], **Bart Jansen**[1,4]

**1** Department of Electronics and Informatics (ETRO), Vrije Universiteit Brussel (VUB), Brussel, Belgium, **2** Faculty of Computing, JiT, Jimma University, Jimma, Ethiopia, **3** Department of Obstetrics and Gynecology, College of Health Science, Jimma University, Jimma, Ethiopia, **4** imec, Leuven, Belgium

\* gtegenaw@vub.be, gelapril1985@gmail.com

## Abstract

### Background

In low-resource settings, patient referral to a hospital is an essential part of the primary health care system. However, there is a paucity of study to explore the challenges and quality of referral coordination and communication.

### Objective

The purpose of this research was to analyze the existing paper-based referral registration logbook for maternal and child health in general and women of reproductive age in particular, to improve referral coordination and evidence-based services in Low-Resource Settings.

### Methods

This study analyzed the existing paper-based referral registration logbook (RRL) and card-sheet to explore the documentation of the referral management process, and the mechanism and quality of referrals between the health center (Jimma Health Center-case, Ethiopia) and the Hospital. A sample of 459 paper-based records from the referral registration logbook were digitized as part of a retrospective observational study. For data preprocessing, visualization, and analysis, we developed a python-based interactive referral clinical pathway tool. The data collection was conducted from August to October 2019. Jimma Health Center's RRL was used to examine how the referral decision was made and what cases were referred to the next level of care. However, the RRL was incomplete and did not contain the expected referral feedback from the hospital. Hence, we defined a new protocol to investigate the quality of referral. We compared the information in the health center's RRL with the medical records in the hospital to which the patients were referred. A total of 201 medical records of referred patients were examined.

**Data Availability Statement:** Due to privacy and ethical issues, data cannot be made public. Data are available upon request to researchers who

meet the requirements for access to sensitive data. Contact information for a data access committee, ethics committee, or other institutional body. Ethical review board (ero@ju.edu.et) or corresponding authors.

**Funding:** The author(s) received no specific funding for this work.

**Competing interests:** The authors have declared that no competing interests exist.

## Results

A total of 459 and 201 RRL records from the health center and the referred hospital, respectively, were analyzed in the study. Out of 459, 86.5% referred cases were between the age of 20 to 30 years. We found that "better patient management", "further patient management", and "further investigation" were the main health-center referral reasons and decisions. It accounted for 40.08%, 39.22%, and 16.34% of all 459 referrals, respectively. The leading and most common referral cases in the health center were long labor, prolonged first and second stage labor, labor or delivery complicated by fetal heart rate anomaly, pre-term newborn, maternal care with breech presentation, premature rupture of membranes, malposition of the uterus, and antepartum hemorrhage. In the hospital RRL and card-sheet, the main referral-in reasons were technical examination, expert advice, further management, and evaluation. We found it overall impossible to match records from the referral log-book in the health center with the patient files in the hospital. Out of 201, only 13.9% of records were perfect matching entries between health center and referred hospital RRL. We found 84%, 14.4%, and 1.6% were appropriate, unnecessary and unknown referrals respectively.

## Conclusion

The paper illustrates the bottlenecks encountered in the quality assessment of the referrals. We analyzed the current status of the referral pathway, existing communications, guidelines and data quality, as a first step towards an end-to-end effective referral coordination and evidence-based referral service. Accessing, monitoring, and tracking the history of referred patients and referral feedback is challenging with the present paper-based referral coordination and communication system. Overall, the referral services were inadequate, and referral feedback was not automatically delivered, causing unnecessary delays.

## 1. Introduction

The World Health Organization (WHO) defines referral as "*a process in which a health worker at one level of the healthcare system, having insufficient resources to manage a clinical condition, seeks the assistance of a better or differently resourced facility at the same or higher level to assist in, or take over the management of, the client's case*" [1]. An effective referral system can play an important role in delivering cost-effective health-care services and ensuring access to better quality care. There is a growing body of literature that recognizes the automation of the paper-based referral system to reduce unnecessary waiting time and delay [2,3], improve the communication gap [4–7], and provide a fast response rate and accurate data [8]. However, in Ethiopia, not all health centers have the required early detection facilities to make sound referral decisions [9,10]. Literature evidence suggests that assisting the referral decision process through decision support systems enables to: (I). aid the process of filling out a referral template [11], (II). provide an alternative recommendation using historical records [12], and (III). prioritize illness severity instead of first-come first-serve rules to ensure on-time treatment of emergency patients [13]. Therefore, there is a need to introduce referral quality improvement techniques beyond looking at the statistical patterns and trends [14]. Clinical pathways (CP) have been introduced for point of care patient management and defined as a concise and

evidence-based summary of the care process, including an algorithm, the informed guidelines or the best evidence, coordinating roles, sequencing activities, documenting and evaluating variance [15,16]. Data from several studies suggest that CP has been successful for minimizing cost, reducing delay, or improving patient outcomes [17–20]. CP helps to bring the whole treatment of a special disease in one concrete setting [21–23]. CP also implemented by tracing historical record and has effective for improving outcomes, reducing delay and cost, as described in [24–28]. In all, clinical decision support in CPs has largely been implemented to manage the quality of the care process (Chawla et al. 2016) [29] but are often out of reach for developing countries. Therefore, it makes sense to develop an automated plan of care helping to practice and enable evidence-based decision-making considering low clinical competence and shortage of resources. It will also empower local and (unexperienced) care providers and enable them to make better referral decisions.

The health network model with three-tiered structure (primary, secondary and tertiary levels) is adopted to structure the Ethiopian healthcare system [30]. Each level is expected to serve thousands of people in the health center to millions of people in specialized hospitals. The referral system is organized based on geographically defined catchment areas. The catchment network links primary hospitals to health centers and the health centers are linked with the health posts (the lowest level of health care service under the health center). The primary hospitals connect with general hospitals and the highest level of network connects general hospitals to specialized hospitals [9,30]. The *primary referral service* is delivered if a specific patient has met the required referral criteria, prerequisites, and the liaison office's approval. **Fig 1** presents the overall summary of the Ethiopian primary healthcare referral flow.

Upon arrival, the health professional will capture the idea (chief compliant), the information, and the feeling of the patient (signs and symptoms) for delivering the required treatment or referral service. A patient can be referred to the next level when a patient needs more inpatient care, expert advice, technical examination, or a technical intervention that is beyond the capabilities of the facility. A referral is also made when the referring facility cannot accept more patients due to a shortage of beds or the unavailability of professionals. In addition, referral from the primary hospital to the health center is also made based on the specific condition of the patient and the prescribed service [30]. Once the referral reason is fulfilled, the liaison office is responsible for: (I). matching, and validating the referral-out,(II). checking that the referral form is filled-in and signed by the physician and ensuring that feedback is sent to the referring health facility, and (II). compiling, analyzing and interpreting data to improve the referral service. The referral side (usually the referred hospital) should confirm the availability of the service, the availability of a bed, and professional readiness. Then, the admitting physicians in the referred hospital are responsible for delivering appropriate diagnosis and evaluation, adhering to the guideline, indicating the level of urgency, ensuring the required procedures and completing all the required documentation. An effective evidence-based service and point-of-care instrument is key for the referral quality and facilitation of seamless referral coordination between the health center (HC) and the referred hospital. Though the referral linkage was documented well, far too little attention has been paid to examine the applicability and implementation challenges in low resource settings that arose from the current paper-based referral registration logbook coordination and communication. Since unnecessary referral will create bottlenecks and heavy patient load in the referred hospital and missed referral will create a delay and complications in the management of the individual patient, a case study on the Ethiopian primary healthcare system (specifically at Jimma Health Center) was conducted to investigate the referral mechanism between the health center and the hospitals, the reasons for the referral, and the "quality" of these referrals. Assessing the quality and completeness of the paper-based records related to the referral system proved to be

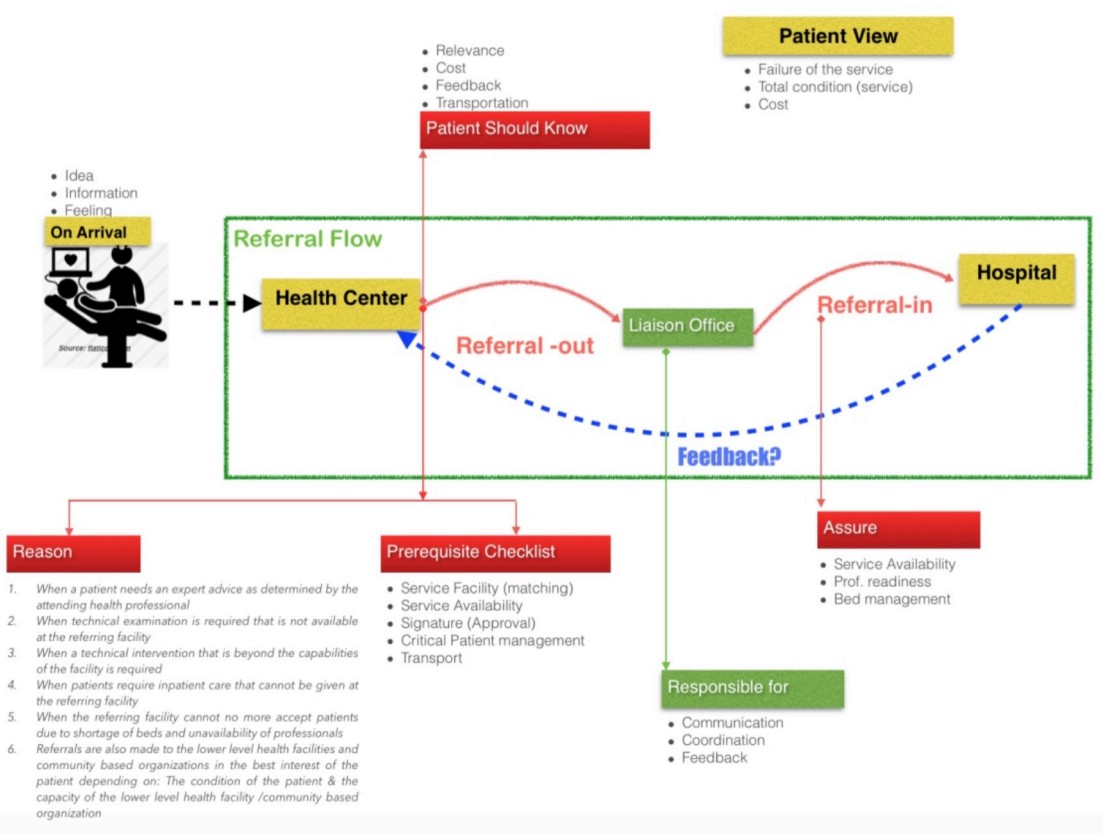

**Fig 1. The primary health care referral flow (FDRE MoH, 2010).**

extremely difficult with respect to completeness, availability of referral feedback and referral reason matching [31]. The aim of this investigation was to examine the existing paper-based referral registration logbook(RRL) and card-sheet to explore the documentation of the referral management process and the mechanism and quality of referrals between the health center (Jimma Health Center-case, Ethiopia) and the Hospital. To address this objective, sample records from the referral registration logbook at the Jimma Health Center and referred hospital were digitized to explore: (i) the reason for referral decision (how the referral decision is made), (ii) what cases are referred to the next level, and (iii) what are the missing feedbacks and conclusions to distinguish appropriate referral from unnecessary referral.

## 2. Methods

### 2.1. Ethical considerations

Ethical clearance was obtained from the Jimma University, Institute of Health, Institutional Review Board (IRB). IHRPGI/467/2019 is the reference number. Permission was granted by the Jimma health center, and the data was gathered and analyzed anonymously after written consent was obtained.

### 2.2. Study setting and design

The research was limited to the exploration of the existing paper-based referral registration logbook for maternal and child health in general and women of reproductive age in particular,

to improve referral coordination and evidence-based services in Low-Resource Settings (LRS). A retrospective observational study in the Ethiopian primary healthcare service, namely at Jimma Health Center (which is in East and Sub-Saharan Africa) was conducted after securing the required ethical clearance and administrative procedure. We collected n = *459* cases from the RRL to investigate our research questions. The number of women of reproductive age who received care at the health center is estimated to be N = 15646. The minimum required sample size was calculated to be n(minimum) = 375. The following formulae were applied:

$$n(minimum) = N*X/(X + N - 1) \hspace{2cm} Eq1$$

$$X = Z_{\alpha/2}{}^2 *p*(1-p)/MOE^2 \hspace{2cm} Eq2$$

in which $Z_{\alpha/2}$ is the critical value of the Normal Distribution at α/2 (confidence level is set to 95% and hence the critical value is 1.96, α is 0.05), the margin of error (MOE) is 0.05, p is the sample proportion and N is the population size. The theory behind these calculations is explained in [32,33]. We follow the recommendations of WHO stepwise approach to surveillance for the remaining values [34]. These value settings allow us to simplify the calculations as formulated in Eqs 1 and 2: p = 0,5 maximizes the nominator in Eq 2 and produces a worst case (i.e. maximal) value for n(minimum), and p = 0,5 is recommended in cases where no a priori results can be used from previous studies; since we only consider women of reproductive age (age 15–48), the number of "age-gender" categories is equal to 1; the response rate is ~100% since we are performing a retrospective analysis based on RRL; the design effect is set equal to 1 which is recommended for random samples, following the WHO guidelines no finite population correction is applied. The representativeness of the data in our sample is guaranteed, since we digitalized RRL records of n = *459* referral consultations, which is larger than n (minimum).

## 2.3. Data source

Clinical guidelines, card-sheet, and referral registration logbook were the main data sources. In health center, the paper-based referral registration logbook (RRL) used to capture documentation about how the referral process was managed, manage the quality of referral using referral feedback to ensure whether the referral was appropriate for the patient or not, manage referral coordination and communication. The RRL used as a standard referral-in and referral-out data entry protocol for managing the referral cases and quality [30]. On average, the health-center paper-based RRL contains 22 records per page and 20–25 pages in total. More information on the structure and layout of the health center paper based RRL is presented in **Fig 2**.

## 2.4. Data collection process

Two data collectors were recruited from the Jimma Health Center to collect the data from the patient referral-out registration logbook and register in the pre-prepared electronic data-sheet template. The data collection was conducted from August to October 2019 and maintained a single data entry process for the duration of the study. We examined the RRL records referred in the year 2010/2011 E.C. (i.e. 2018/19 G.C). Sample records from the referral registration logbook at the Jimma Health Center and referred hospital were digitized. **Table 1** presents the summary of the dataset.

To assure the quality of data, the data collectors were recruited based on their familiarity and experience with the existing workflow process, patient card-sheet management, professional expertise, and exposure to handle clinical and health information. The data collectors

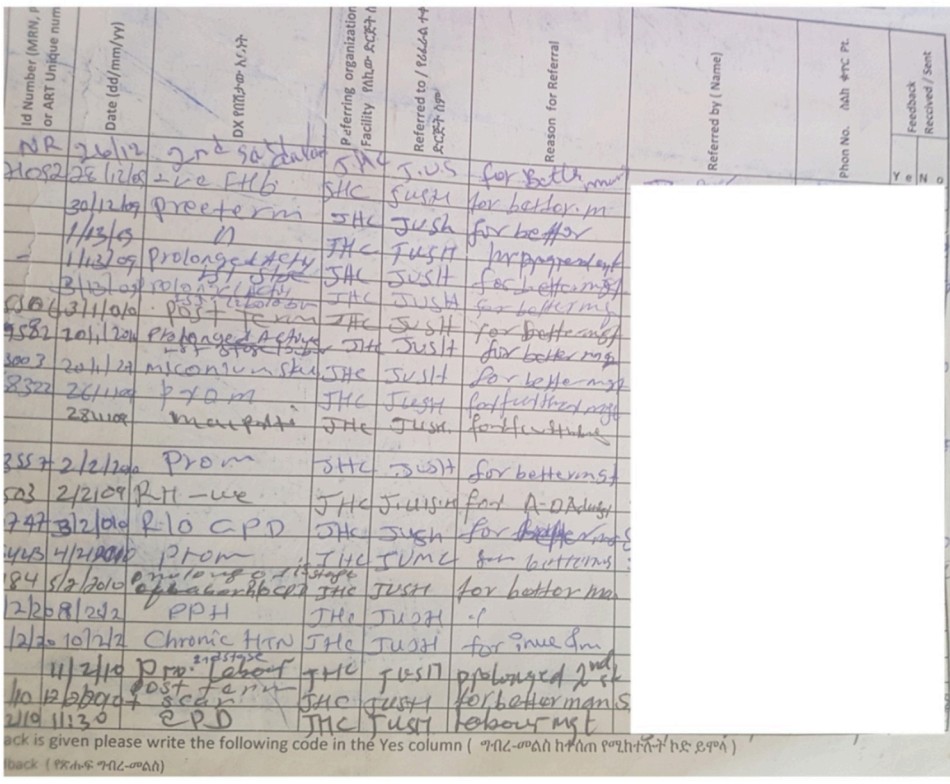

**Fig 2. The paper-based health center RRL.**

were already working as full-timers (health information system, data monitoring and recorder professional) and they agreed to cooperate on the data collection process during the weekend and after working hours.

However, the health center referral logbook did not have the expected referral feedback from the hospital. To get the referral feedback, we defined a new strategy and recruited two additional data collectors to collect from the referred hospital logbook and card-sheet. The referred hospital RRL data was also collected using a pre-prepared electronic data-sheet template. The overall data collection process was random and obtaining the required information was time-consuming and tougher than expected. Then, we tried to compare the information in the referral logbook of the health center with the medical records in the hospital to which the patients were referred. Though the health-center cases were referred to Jimma University Specialized Hospital and Shanan Gibe General Hospital, this study was limited to extract referral feedback from Jimma University Specialized Hospital.

Finally, the alignment of the CP dataset classification based on International Classification of Diseases 10th Revision (ICD-10) standards was performed [35].

**Table 1. Summary of the health center and referred hospital RRL dataset.**

| No. of Cases | Data Sources | Status |
|---|---|---|
| 459 | Jimma Health Center RRL | Without referral feedback |
| 201 | Jimma Referred Hospital Card-sheet and RRL | With proper and written feedback |

## 2.5. Data preprocessing and analysis

Data pre-processing, visual inspection and analysis of the CP dataset were conducted using a python based interactive referral CP visualization tool. An interactive and RRL data-driven tool was designed for: (I). Analyzing and visualizing RRL records, (II). Pre-processing and visual inspection of RRL records such as handling missing, noisy and outlier values, and (III). Conducting automated referral feedback and reasoning analysis.

To answer our research questions, all the records were collected from the Jimma Health Center RRL and the referred hospital (i.e. Jimma University Specialized Hospital) registration logbook and card-sheet. Since all the records were presented in hard-copy format this activity was time-consuming, and it was difficult to extract the right and perfect matching records.

WHO data-quality review and metrics were adopted to assess the quality of RRL [36]. The referral quality was assessed with respect to: (I). The completeness and timeliness of RRL data, (II). The internal consistency of RRL such as the presence of anomalies, the RRL format and type, and consistency over time (history of recording), (III). The accuracy, validity, and uniqueness of RRL data to describe the RRL definition, RRL column type, RRL column value, and level of duplication, and (IV). The external consistency such as the matching of health-center referral reasoning and referred hospital referral feedback.

The RRL analysis for the health-center was then conducted. The RRL contains information on patient referrals such as "disease name (cases)," "referred to," "referral reason," "referred date," and "collected signs and symptoms" that were used in primary care prior to making a referral decision. We examined the reasons for referral, disease names (cases), and conclusion (outcome) to see the reasons for the referral decision and which cases were referred to the next level. The referral justification was registered in the RRL "referral-out reasoning" column and documented based on the national guidelines (or referral templates). In order to assess referral cases, it is crucial to examine this referral-out justification (reasons). Fig 1 gives further information about the reason for the referral (the decision to refer or not to refer a patient) based on the national guidelines, which should be medically sound, objective, and in the best interests of the patient.

The referred hospital RRL, on the other hand, was analyzed to identify the missing feedbacks and conclusions to distinguish appropriate referral from unnecessary referral. Examining the RRL column "the final remark and feedback," which was filled by the referring hospital and forwarded to the health center for future referral reference and quality improvement, helped determine the appropriateness of the referral. The health-center referral reason, hospital referral feedback, and conclusion were all analyzed to examine which referrals were appropriate and which were not. We also compared the RRLs from the health center and the referring hospital to ensure that the datasets (or individual records) were matched in general and that the referral reasoning was accurate in particular.

The final CP dataset contains medical record number, disease name (cases), age, sex, referring organization, referred to, referral reason, referred date, date of arrival at the hospital, collected signs and symptoms, referral feedback and conclusion (outcome), International Classification of Diseases 10th Revision (ICD-10) Classification, ICD-10 parent category, matching terms and descriptions attributes. Overall, we analyzed the health center RRL for investigating the referral case and reasons, and the referred hospital RRL for exploring the missed referral feedback. Comparison between the health center and referred hospital RRL were also conducted to verify the matching of the datasets and referral reasoning.

## 3. Results

A total of 459 cases were collected from the Jimma Health Center's RRL. The Jimma Health Center refers patients to the Jimma University Specialized Hospital and Shanan Gibe General

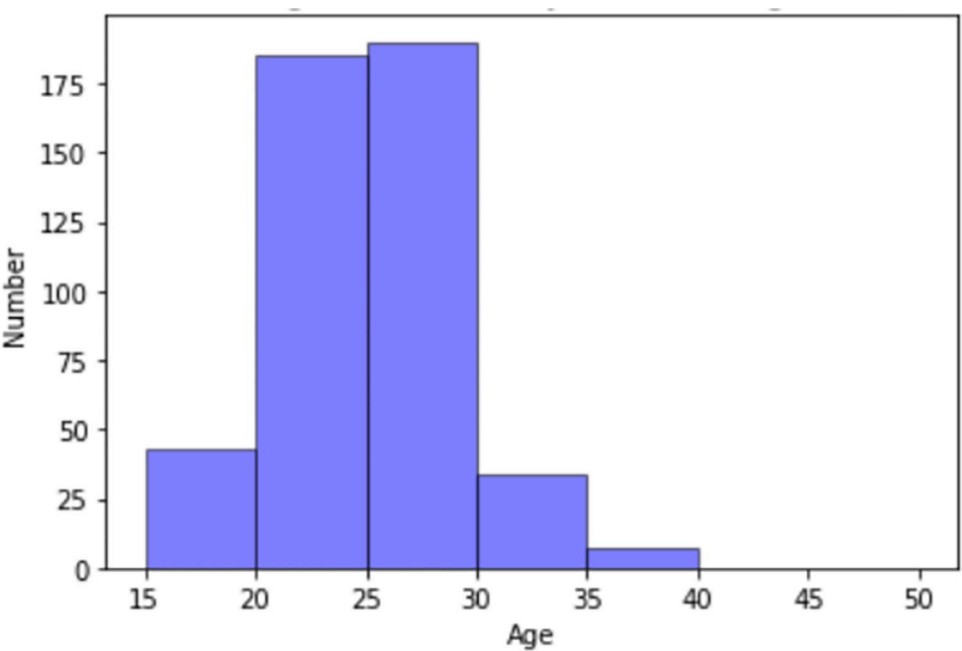

**Fig 3. Summary of referral age.**

Hospital. Out of 459, 86.5% referred cases were between the age of 20 to 30 years. **Fig 3** is depicted to illustrate the summary of referral age.

The finding regarding RRL data quality metrics, indicate that all RRL columns were documented except "referral feedback" column. However, the "referral feedback" column is consistently empty. A detailed analysis of RRL data quality metrics is presented in **Table 2**.

The referral decision (the reason for the referral) is made based on the signs and symptoms collected at the RRL health center in accordance with the diseases categories (or cases). The collected signs and symptoms were used to assess the risk of diseases (cases), and the referring health professional documented the referral-out justification. We found that meeting any of the six referral criteria (the reason for the referral) influenced the decision to refer based on the referral guidelines, which include *"When a patient needs an expert advice as determined by the attending health professional", "When technical examination is required that is not available at the referring facility", "When a technical intervention that is beyond the capabilities of the facility is required","When patients require inpatient care that cannot be given at the referring facility", "When the referring facility cannot no more accept patients due to shortage of beds and unavailability of professionals", and "Referrals are also made to the lower level health facilities and community based organizations in the best interest of the patient depending on the condition of the patient and the capacity of the lower level health facility /community based organization"* [30].

Using health-center RRL categories such as disease name (or cases), signs and symptoms, conclusion, and referral-out reason, we examined the referral cases and the reasons for the referral decisions. These findings suggest that long labour, prolonged first and second stage labour, labour or delivery complicated by fetal heart rate anomaly, preterm newborn, maternal care with breech presentation, premature rupture of membranes, malposition of the uterus and antepartum hemorrhage were the leading and most frequent referral cases in the health center. We found that "better patient management", "further patient management", and "further investigation" were the main health center referral reasons. It accounted for 40.08%,

**Table 2. Analysis of the RRL data quality metrics.**

| Data Quality Metrics | | Result Found |
|---|---|---|
| RRL column format and type | | • The overall RRL was recorded according to the RRL column format and type. There were three columns with a value and format of date and time, numeric, and categorical features. Whereas, the remaining RRL columns were string value and format. |
| Accuracy and uniqueness | | • All the RRL column attribute values were valid and it conforms to the syntax (format and type) of the RRL definition.<br>• We observed that the RRL information reflects how the referral process coordination and communication were managed<br>• None of the RRL values were duplicated. |
| Completeness | | • Of 459, 100% of all RRL columns were documented except the "referral feedback" columns. The "referral feedback" column was consistently empty (0%).<br>• Among the RRL "referral date" column value, 98% were non-missing values. Only 2% of the referral date were not recorded. |
| Timeliness | | • The RRL column records were found to be up to date and reported as such. Referral feedback columns, on the other hand, were usually missing and were not used for real-time reporting. |
| Consistency and validity | Internal consistency of RRL | • Anomalies were found. For example, two age values (i.e. 88 and 2423) replaced by the average value of the age because the two values were considered as a typing error.<br>• The layout and format of the handwritten information presented in the patient card-sheet and RRL were inconsistent from one record to another<br>• Health center RRL reasons did not match the national guidelines. |
| | External consistency between Health-center RRL-out and Hospital RRL-in | • We found it overall impossible to match referral reasoning from the referral logbook in the health center with the referral feedback in the hospital.<br>• The health center RRL reason was not exactly followed by the hospital. |

39.22%, and 16.34% of all 459 referrals, respectively. *Further patient management* indicates that there are some findings in the health center, but more experienced physicians should do extra examinations and investigations whereas in the case of *better patient management* there are not enough results in the health care center to confirm the patients' diagnosis. In addition, normal delivery was referred to Shanan Gibe General Hospital with a referral reason of lack of light (electricity breakdown), prolonged 2$^{nd}$ stage labor, and high blood pressure. The detailed summary of the health center referral reason is presented in **Fig 4**.

We found that there was no evidence in the health-center RRL to distinguish appropriate referrals from unnecessary referrals. The health center's referral logbook includes the required column for "referral feedback" but it is consistently empty. The next section of the study was concerned with investigating the quality of referral (or referral feedback) from the referred hospital and a total of 201 cases were examined. We found it overall impossible to match referral reasoning from the referral logbook in the health center with the referral feedback in the hospital. Out of 201, 13.9,18.9, and 67.2% of records were perfect matching, a high probability of matching with educated guesses, and no clue who-is-who respectively. We observed that the referral reasons in the hospital record were explicit and more explanatory in comparison with the HC's RRL referral reason. Out of 201 cases, 84.0,14.4, and 1.6% were appropriate referrals

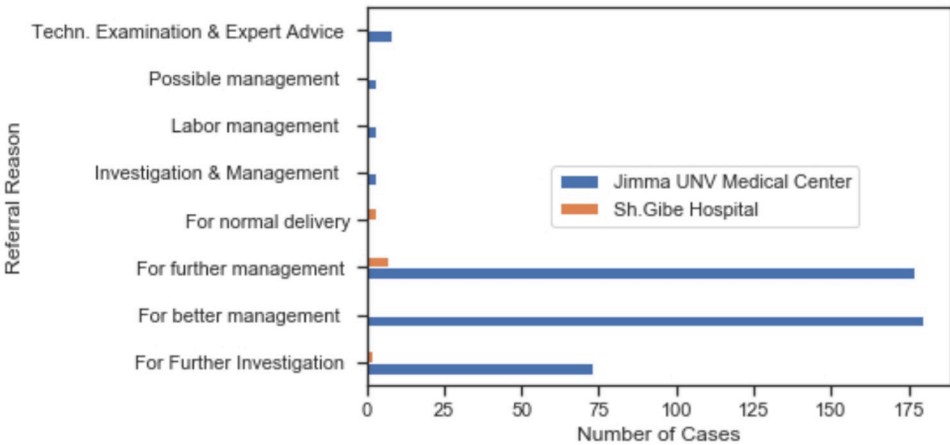

**Fig 4. Summary of the health center referral reason.**

(the referral was right, and this is encouraging to refer similar cases in the future), unnecessary referrals and not-known respectively. The criterion for an appropriate referral was extracted from the hospital RRL (feedback and conclusion section filled-in by the physician). As mentioned before, further management and evaluation, investigation and better management were the leading referral reasons next to technical examination and expert advice. There was unnecessary referral in cases of a referral reason for technical examination and expert advice, further management and evaluation, better medical prescriptions, investigation, and Mx. (medication management). The summary of the result is presented in **Fig 5**.

Further analysis showed that "further investigation", "further management" and "better management" were the leading referral-out reasons, whereas technical examination, expert advice, further management and evaluation were the main referral-in reasons in the hospital RRL, as mentioned in the hospital card-sheets. **Fig 6** presents the detailed aggregated summary

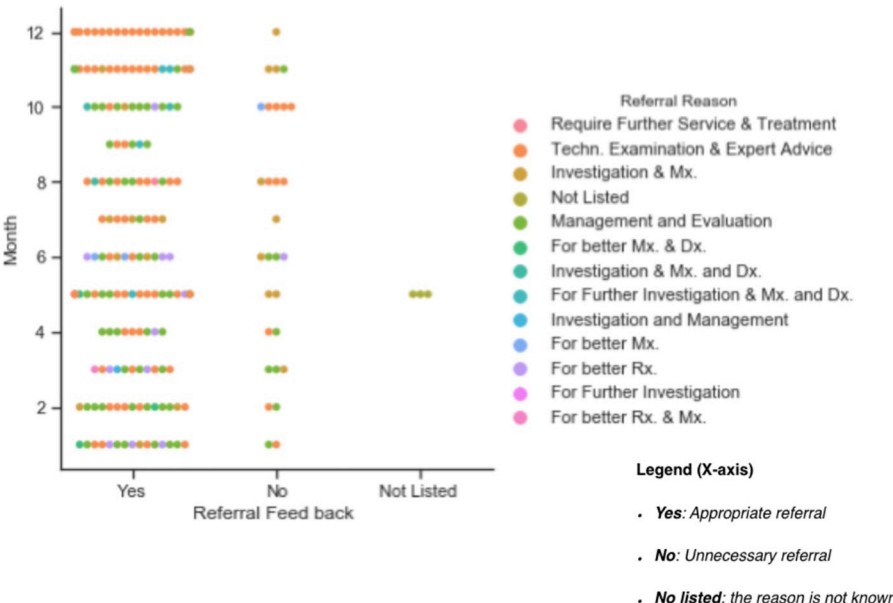

**Fig 5. Summary of referral feedback based on the hospital RRL.**

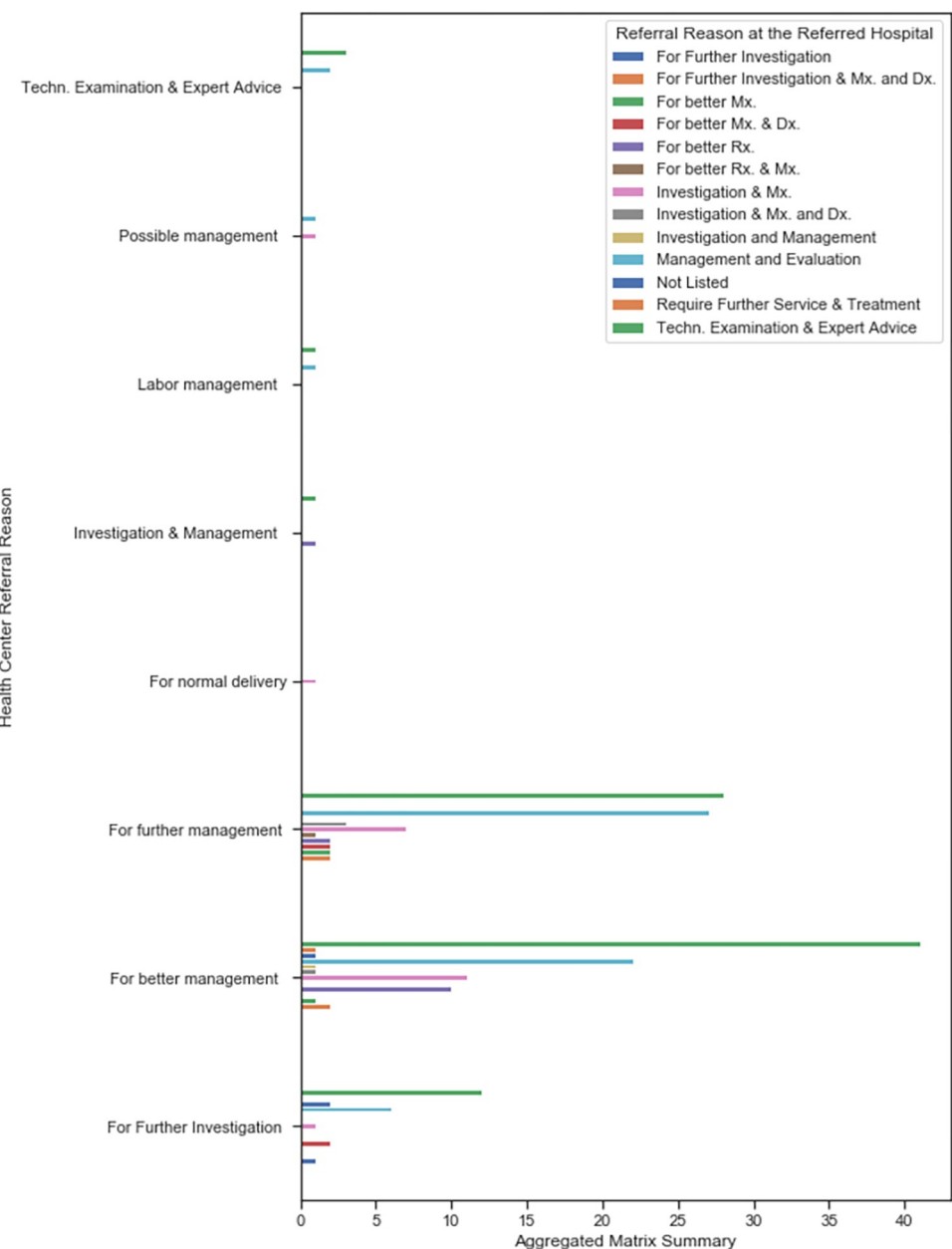

**Fig 6. Aggregated matrix summary be referral reason compute based on HC RRL_out and Hospital RRL_in.**

of referral reason mentioned in the heath-center RRL and the hospital RRL. We also noticed that the registration of health-center RRL referral reasons did not exactly match and follow the national guidelines.

Our finding revealed that it is challenging (almost impossible) to satisfy and deliver the expected referral feedback with the current paper-based referral coordination. In summary, these results show that: (I). All referrals were made from the health-care center to the hospital, (II). The health-center "referral feedback" RRL column was incomplete and empty, (III). The leading referral reasons were "better patient management", "further patient management", and "further investigation" respectively, and (IV). Getting referral feedback from the referred

hospital was very challenging and difficult to match health-center referral reasoning and referred hospital referral feedback.

## 4. Discussion

This study was set up to assess the referral decision and quality of referrals between the health center (Jimma Health Center) and the hospital (Jimma University Specialized Hospital). In accordance with the present results, 11/12 (91.7%) columns were filled which met the national guidelines expected target of 2019/20 (i.e. 90% of data completeness). With respect to the national health sector transformation plan, the utilization of evidence-based information for decision-making was not known and only 29% of the health facilities met the minimum information standard (criteria) whereas report completeness and timeliness was 72% and 84% respectively [9].

The referring health professional makes the decision and fills out the RRL refer-out reason based on the referral criteria. Prior to making a referral decision, the referring health professional is responsible for filling out the RRL form with the essential information and attaching any relevant documents. The RRL contains information on patient referrals such as "disease name (cases)," "referred to", "referral reason", and "collected signs and symptoms" that were used in primary care. RRL categories "signs and symptoms" and "diseases (or cases)" were used to make the referral decision, as well as the conclusion (or outcome). The signs and symptoms were used to assess the risk of diseases (cases), and the referring health professional documented the referral-out justification. Referral decisions are made when the criteria for referral are medically appropriate and in the best interests of the patient or client. Furthermore, the national guidelines are also used as a reference guide to address the most common presenting symptoms and referral criteria, which aids in the prioritization and documentation of the referral condition [30,37]. It is somewhat surprising that the registration of health-center RRL referral reason did not exactly match and follow the national guidelines. For instance, the guideline set referral reason as "A *patient needs an expert advice as determined by the attending heath professional*", "*Technical examination is required that is not available at the referring facility*", "*Technical intervention that is beyond the capabilities of the facility is required*" and so on whereas the referral reason written in the RRL looks like "*better patient management*", "*further patient management*", "*further investigation*" and so on. A possible explanation for this might be that the benefit of time savings due to the unnecessity of data transcription and the layout of the paper-based RRL not enough to accommodate the national guideline referral reasoning.

In the health-center RRL, the "referral feedback" column that helps to mitigate unnecessary referral, missed referral (or referral delay), and complications in the management of the individual patient were missed. The referral feedback is crucial to verify whether the referral case is correct or not. The final remark and feedback were sent from the referred hospital to the health center for future referral reference and quality improvement. Missing timely referral feedback and evidence will affect the quality of the healthcare service in the health center and the staff cannot know whether their referral policy is of good quality. When the patient who was referred to hospital show up for subsequent follow-up in the health center, surprisingly the health center has no information on what happened in the hospital. Hence, they often consider this as a new case for the patient. With the referral logbook being incomplete it is impossible to answer the initial research questions on the quality of referral. Therefore, as mentioned earlier, we defined a new approach to investigate the quality of referral and tried to compare the information in the referral logbook of the health center with the medical records in the hospital to which the patients were referred. However, the observed referral reasoning difference between the referral logbook in the health center with the referral feedback in the hospital was

significant. The difference of referral reason between health center RRL and hospital RRL may arise from:(i) the limitation of the dataset (for instance, the proportion of matching entries on referral between health center and referred hospital RRL was 13.9% only), (ii) lack of proper written documentation and digitization, and (iii) the level of expertise, availability of resources and level of investigation.

This finding suggested that it is very challenging to access, monitor, and track the history of the referral patients and referral feedback using current paper-based referral coordination and communication. There are several possible explanations for this result: (I). The lack of adequate communication between the primary care and the referred hospital, (II). The lack of adequate infrastructure and integrated platform with the existing workflow to enable automated referral system, (III). The layout and format of the handwritten information presented in the patient card-sheet and RRL were inconsistent from one record to another, (IV). The referring provider did not automatically notice the missed referral feedback. This observation may support the hypothesis, enabling an automated referral system may improve the communication gap, and reduce waiting time between the primary health center and the referred hospitals [6]. The evidence-based point-of-care instrument can thus be suggested that assist the process of filling out a referral template to reduce time and error [11] and helps to assist the referral follow-up [38]. However, further investigation is required to explore the barriers and facilitators for enhancing the quality of referral service and to deliver a fully functional and effective referral system.

Overall, these findings raise intriguing questions regarding the nature and extent of the missed referral feedback, the quality of referrals, and referral decisions. Furthermore, the RRL analysis showed that all referrals are made from the health-care center to the hospital. Unfortunately, we did not find a referral made to lower-level health facilities.

Our study has several limitations. We sampled from a Jimma Health-center and focused on women of reproductive age and limited to extract referral feedback from Jimma Specialized Hospital. Our findings thus may not easily generalize to other health-centers. Moreover, this study examined the bottlenecks encountered in the quality assessment of the referrals with current paper-based documentation, coordination, communication, and patient files.

## 5. Conclusions

The study addressed the question of how the referral decision (the reason for referral decision), coordination, and communication are made based on the collected dataset from the health center RRL and referred hospital RRL. It uncovers the missed referral feedback using the referred hospital RRL. With the current paper-based referral coordination and communication, it is very challenging to access, monitor and track the history of the referred patients and referral feedback. We also observed that the health center RRL reason was not exactly followed by the hospital and does not match the national guidelines. The most crucial problem was in the identification of referral records (no unique, consistent and easy searchable identifier was present in the RRLs and card-warehouse). The layout and format of the handwritten information presented in the patient card-sheet and RRL were inconsistent from one record to another, and hence it was difficult to audit the records. In addition, if changes were made, it was not easy to track where the changes were made. Because of all these, the quality of the referral services in primary healthcare settings was compromised, referral feedback was not delivered automatically, and this caused unnecessary delay. Therefore, when the necessary infrastructure and human resources are in place, implementing an automated patient management system may help *to correct and impose a standardized referral service workflow*. It creates the opportunity to track and monitor the referred patients, access their medical records,

retrieve referral feedbacks, and standardize the interaction between the patients and health professionals. As mentioned before, implementing a patient data (or smart) card seems to be the fastest solution to improve the referral service quality in the absence of distributed digital infrastructure in such low-resource setting. In summary, implementing digital referral services compatible with the existing clinical guidelines is urgently needed for improving and assessing the quality of referral services in low-resource settings. This is the subject of our further research. Furthermore, even before viable infrastructure is implemented and competent human resources are attracted, a minimal infrastructure (a low-cost alternative) and trained personnel can facilitate referral coordination and offer evidence-based services to ensure a buy-in and acceptance by all stakeholders. This transition needs careful planning and openness to adopt best practices in order to reduce resource waste, to ensure training, maintenance, and avoid heavy service expenses.

## Acknowledgments

The NASCERE (Network for Advancement of Sustainable Capacity in Education and Research in Ethiopia) program has assisted us in the work to date and will continue to assist us as we move forward with the planned activities. Besides NASCERE, we also acknowledge the efforts of Dr. Bitiya Admasu, Dr. Kume Bekele, Dr. Rediet, and Dr. Gizat Molla who volunteered to deliver their feedback and comments on the collected dataset and documents. We are very grateful towards all the data col- lectors and the card store officer—special thanks to Mr. Bantewesn (Hospital data manager and facilitator) and Mr. Sultan (health center HIMS expert).

## Author Contributions

**Conceptualization:** Geletaw Sahle Tegenaw, Frank Verbeke, Jan Cornelis, Bart Jansen.

**Data curation:** Geletaw Sahle Tegenaw.

**Formal analysis:** Geletaw Sahle Tegenaw.

**Investigation:** Geletaw Sahle Tegenaw.

**Methodology:** Geletaw Sahle Tegenaw.

**Software:** Geletaw Sahle Tegenaw.

**Supervision:** Demisew Amenu, Girum Ketema, Frank Verbeke, Jan Cornelis, Bart Jansen.

**Validation:** Geletaw Sahle Tegenaw.

**Visualization:** Geletaw Sahle Tegenaw.

**Writing – original draft:** Geletaw Sahle Tegenaw.

**Writing – review & editing:** Geletaw Sahle Tegenaw, Frank Verbeke, Jan Cornelis, Bart Jansen.

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
