## [Decision Letter · Decision Letter 0]

1 Dec 2021

PONE-D-21-20872Analysis of referral pathways in Low-Resource Settings for enabling an evidence-based point-of-care instrumentPLOS ONE

Dear Dr. Tegenaw,

Thank you for submitting your manuscript to PLOS ONE. After careful consideration, we feel that it has merit but does not fully meet PLOS ONE’s publication criteria as it currently stands. Therefore, we invite you to submit a revised version of the manuscript that addresses the points raised during the review process.

The reviewers raised a number of issues with the study. These included questions about the methodological approach, clarification on the sample size and the presentation of results. Their comments are summarized below and can be viewed in detail in the attached files.

We look forward to receiving your revised manuscript.

Kind regards,

Natasha McDonald, PhD

Associate Editor

PLOS ONE

Journal Requirements:

2. As per PLOS ONE's data sharing policy, PLOS ONE does not allow for censored information. As such please revise your manuscript accordingly and provide the names of the participating sites of your study.

5. We suggest you thoroughly copyedit your manuscript for language usage, spelling, and grammar. If you do not know anyone who can help you do this, you may wish to consider employing a professional scientific editing service. Whilst you ma We suggest you thoroughly copyedit your manuscript for language usage, spelling, and grammar. If you do not know anyone who can help you do this, you may wish to consider employing a professional scientific editing service.

A clean copy of the edited manuscript (uploaded as the new *manuscript* file)y use any professional scientific editing service of your choice, PLOS has partnered with both American Journal Experts (AJE) and Editage to provide discounted services to PLOS authors. Both organizations have experience helping authors meet PLOS guidelines and can provide language editing, translation, manuscript formatting, and figure formatting to ensure your manuscript meets our submission guidelines. To take advantage of our partnership with AJE, visit the AJE website (http://learn.aje.com/plos/) for a 15% discount off AJE services. To take advantage of our partnership with Editage, visit the Editage website (www.editage.com) and enter referral code PLOSEDIT for a 15% discount off Editage services. If the PLOS editorial team finds any language issues in text that either AJE or Editage has edited, the service provider will re-edit the text for free. Upon resubmission, please provide the following: The name of the colleague or the details of the professional service that edited your manuscript A copy of your manuscript showing your changes by either highlighting them or using track changes (uploaded as a *supporting information* file) A clean copy of the edited manuscript (uploaded as the new *manuscript* file)

Reviewers' comments:

Reviewer's Responses to Questions

**Comments to the Author**

1. Is the manuscript technically sound, and do the data support the conclusions?

Reviewer #1: Partly

Reviewer #2: Partly

2. Has the statistical analysis been performed appropriately and rigorously? 

Reviewer #1: No

Reviewer #2: No

3. Have the authors made all data underlying the findings in their manuscript fully available?

Reviewer #1: No

Reviewer #2: Yes

4. Is the manuscript presented in an intelligible fashion and written in standard English?

Reviewer #1: Yes

Reviewer #2: Yes

5. Review Comments to the Author

Reviewer #1: REVIEWERS COMMENTS

ABSTRACT

In the Abstract section the results indicate that out of 201. It is not clear what the N(number of matching samples are that lead to the 13%). Kindly indicate this. The results of the study are not compatible with the objectives. The objectives indicate that the study intended to explore the practice whereas the results indicate the main reasons for referral. Kindly align the two. The conclusion section should indicate the overall implication of the study within the health system. It currently reads like an objective. Kindly revise to indicate the implication.

DATA COLLECTION SECTION

In page 11 you mention that 459 cases were used. It is not clear how the sampling was done. As much as the authors indicate the stepwise approach the calculation that is present in the paragraph below does not match the sample size given. Kindly clarify.

DISCUSSION SECTION

In Page 16 you indicate that the proportion of “matching”. The sentence seems incomplete. Is it matching entries on referral?

In the second paragraph in the discussion section you mention that the reasons for referral do not match the guidelines. In my view – the intention is not to exactly match the guidelines word for word. Whereas the reasons for referral in the RRL book indicate reasons that you mention such as “better patient management” they convey the nuance of the needs of the healthcare worker and should hence not be seen as a radical departure. The guidelines should be seen as guidelines and not an absolute way of filling in a referral form. I suggest that you discuss the evidence on the practical use of the form as a way of describing the departure from the guidelines.

In the fourth paragraph pg. 16 revise spelling for found. The sentence on the limitations also needs to start in its own paragraph.

Reviewer #2: Justification must be made to how data was generated and analysed. The various statistical levels the data was subjected to must be explicitly stated. Referencing in the entire write-up should be worked on. I came across Harvard referencing style at some points, APA and MLA especially with the endnote referencing.

6. PLOS authors have the option to publish the peer review history of their article (what does this mean?). If published, this will include your full peer review and any attached files.

Reviewer #1: **Yes: **Jackline Oluoch-Aridi

Reviewer #2: No

---

## [Author Response · Author response to Decision Letter 0]

1 Feb 2022

Rebuttal letter

Dear Editor, 

PLOS ONE Journal 

Website: https://journals.plos.org/plosone/s/journal-information

Dec 27, 2021.

Dear Sir/Madam

Thank you for allowing us the opportunity to submit a revised draft of the manuscript. We thank you and the reviewers for a thorough reading and constructive criticism of our manuscript and for the opportunity to revise and resubmit. We are pleased to submit the improved research article, including a proposed slight title change, “Analysis of referral pathways between low resource health settings to improve coordination and evidence-based services for maternal and child health in Ethiopia” for your consideration in the PLOS ONE. On the following pages, you will find our response to the editor and reviewer comments.

On behalf of my co-authors, I thank you for your consideration of this resubmission. We appreciate your time and look forward to your response.

Please see below, in blue, for a point-by-point response to the reviewers. All page numbers refer to the revised manuscript file with tracked changes. All modifications in the manuscript have been highlighted in red.

Sincerely, 

Geletaw Sahle Tegenaw 

   Title: 

The title should include person and time.

The title of the article does not fully cover the objective of the research. For instance, there is no point in time in the research findings where the care of patients was analysed using the information on Referral Registration Logbook (RRL) as suggested by the title as an evidence-based point-of-care instrument. We appreciate the efforts by you and the reviewers on the manuscript. To better frame the main thesis of our paper, we updated the title to “ Analysis of referral pathways between low resource health settings to improve coordination and evidence-based services for maternal and child health in Ethiopia”

Abstract: 

Reviewer 1: From page 8 under the heading Objectives line 1-2. The objective of the study suggests that the analysis is to enable data-driven clinical pathway but the discussion and conclusion of the research largely focused on the completeness and feedback mechanism of the RRL and card-sheet. Reviewer 2: In the Abstract section the results indicate that out of 201. It is not clear what the N(number of matching samples are that lead to the 13%). Kindly indicate this. The results of the study are not compatible with the objectives. The objectives indicate that the study intended to explore the practice whereas the results indicate the main reasons for referral. Kindly align the two. The conclusion section should indicate the overall implication of the study within the health system. It currently reads like an objective. Kindly revise to indicate the implication.  Thanks for the reviewers’ comment and we appreciate your suggestion. We updated the Methods, Results and Conclusion of the abstract for clarification.  Added and updated as: 

 Methods This study analyzed the existing paper-based referral registration logbook (RRL) and card-sheet to explore the documentation of the referral management process, and the mechanism and quality of referrals between the health center (X Health Center-case, Ethiopia) and the Hospital. (X anonymized for blind review). A sample of 459 paper-based records from the referral registration logbook were digitized as part of a retrospective observational study. For data preprocessing, visualization, and analysis, we developed a python-based interactive referral clinical pathway tool. The data collection was conducted from August to October 2019. X Health Center’s RRL was used to examine how the referral decision was made and what cases were referred to the next level of care. However, the RRL was incomplete and did not contain the expected referral feedback from the hospital. Hence, we defined a new protocol to investigate the quality of referral. We compared the information in the health center’s RRL with the medical records in the hospital to which the patients were referred. A total of 201 medical records of referred patients were examined.   Results A total of 459 and 201 RRL records from the health center and the referred hospital, respectively, were analyzed in the study. Out of 459, 86.5% referred cases were between the age of 20 to 30 years. We found that “better patient management”, “further patient management”, and “further investigation” were the main health-center referral reasons and decisions. It accounted for 40.08 %, 39.22 %, and 16.34 % of all 459 referrals, respectively. The leading and most common referral cases in the health center were long labor, prolonged first and second stage labor, labor or delivery complicated by fetal heart rate anomaly, preterm newborn, maternal care with breech presentation, premature rupture of membranes, malposition of the uterus, and antepartum hemorrhage. In the hospital RRL and card-sheet, the main referral-in reasons were technical examination, expert advice, further management, and evaluation. We found it overall impossible to match records from the referral logbook in the health center with the patient files in the hospital. Out of 201, only 13.9% of records were perfect matching entries between health center and referred hospital RRL. We found 84%, 14.4%, and 1.6% were appropriate, unnecessary and unknown referrals respectively.

 Conclusion The paper illustrates the bottlenecks encountered in the quality assessment of the referrals. We analyzed the current status of the referral pathway, existing communications, guidelines and data quality, as a first step towards an end-to-end effective referral coordination and evidence-based referral service. Accessing, monitoring, and tracking the history of referred patients and referral feedback is challenging with the present paper-based referral coordination and communication system. Overall, the referral services were inadequate, and referral feedback was not automatically delivered, causing unnecessary delays.

Paragraph 3 of page 8 under the methods portion does not indicate the study type used for the research

. We updated the methods and added the study types such as a retrospective observational study. We used this terminology consistently in the text (e.g. also in Section 2.1) 

Methods:

In page 11 you mention that 459 cases were used. It is not clear how the sampling was done. As much as the authors indicate the stepwise approach the calculation that is present in the paragraph below does not match the sample size given. Kindly clarify.

  We appreciate your suggestion and n(minimum)=375 was computed as the minimal required sample size. The data in our sample is guaranteed to be representative since we digitalized RRL records from n = 459 referral consultations.

   We added the following explanation based on the suggestion, in Section 2.1. We collected n=459 cases from the RRL to investigate our research questions. The number of women of reproductive age who received care at the health center is estimated to be N=15646. The minimum required sample size was calculated to be n(minimum)=375. The following formulae were applied: 

n(minimum) = N*X / (X + N – 1) equation 1 X = Zα/22 *p*(1-p) / MOE2 equation 2  in which Zα/2 is the critical value of the Normal Distribution at α/2 (confidence level is set to 95% and hence the critical value is 1.96, α is 0.05), the margin of error (MOE) is 0.05, p is the sample proportion and N is the population size. The theory behind these calculations is explained in (Israel 1992; Daniel and Cross 2018). We follow the recommendations of WHO stepwise approach to surveillance for the remaining values (WHO 2017b). These value settings allow us to simplify the calculations as formulated in equations 1 and 2: p = 0,5 maximizes the nominator in equation 2 and produces a worst case (i.e. maximal) value for n(minimum), and p = 0,5 is recommended in cases where no a priori results can be used from previous studies; since we only consider women of reproductive age (age 15-48), the number of “age-gender” categories is equal to 1; the response rate is ~100 % since we are performing a retrospective analysis based on RRL; the design effect is set equal to 1 which is recommended for random samples, following the WHO guidelines no finite population correction is applied. The representativeness of the data in our sample is guaranteed, since we digitalized RRL records of n = 459 referral consultations, which is larger than n(minimum).

Page 12 paragraph 1 line 3, the study design used for research is quoted as “a case study” but every evidence about how the data was obtained and analyzed indicated that this was a retrospective observational study design. We appreciate that the reviewer asked for more accurate wording regarding the study types. We have substituted “a case study” to “a retrospective observational study”, which was employed in our study (e.g. in the modified text of Section 2.1)

Also, in page 11 paragraph 2, there was no mention of the tool that was used to carry out the study.   Added in Section 2.2: “The referred hospital RRL data was also collected using a pre-prepared electronic data-sheet template.”

In page 12 paragraph 3, line 2 – 5 the analysis was overly concentrated on the completeness and feedback of the referrals but did not cover the other objectives of the research.  Thank you for the comment. We have added the following relevant information for clarification, in Section 2.4.   The RRL analysis for the health-center was then conducted. The RRL contains information on patient referrals such as "disease name (cases)," "referred to," "referral reason," "referred date," and "collected signs and symptoms" that were used in primary care prior to making a referral decision. We examined the reasons for referral, disease names (cases), and conclusion (outcome) to see the reasons for the referral decision and which cases were referred to the next level. The referral justification was registered in the RRL “referral-out reasoning” column and documented based on the national guidelines (or referral templates). In order to assess referral cases, it is crucial to examine this referral-out justification (reasons). Figure 1 gives further information about the reason for the referral based on the national guidelines, which should be medically sound, objective, and in the best interests of the patient.   The referred hospital RRL, on the other hand, was analyzed to identify the missing feedbacks and conclusions to distinguish appropriate referral from unnecessary referral. Examining the RRL column "the final remark and feedback," which was filled by the referring hospital and forwarded to the health center for future referral reference and quality improvement, helped determine the appropriateness of the referral. The health-center referral reason, hospital referral feedback, and conclusion were all analyzed to examine which referrals were appropriate and which were not. We also compared the RRLs from the health center and the referring hospital to ensure that the datasets (or individual records) were matched in general and that the referral reasoning was accurate in particular.  

Results:

In page 13 paragraph 3, the study results are well-represented but this does not look at the issues raised in the objectives which include the decision to refer or not to refer a client (to determine unnecessary referrals).  We appreciate the motivation for these comments. We have clarified, and added the following further clarification.   Added: The referral decision (the reason for the referral) is made based on the signs and symptoms collected at the RRL health center in accordance with the diseases categories (or cases). The collected signs and symptoms were used to assess the risk of diseases (cases), and the referring health professional documented the referral-out justification. We found that meeting any of the six referral criteria (the reason for the referral) influenced the decision to refer based on the referral guidelines, which include “When a patient needs an expert advice as determined by the attending health professional”, “When technical examination is required that is not available at the referring facility”, “When a technical intervention that is beyond the capabilities of the facility is required”, “When patients require inpatient care that cannot be given at the referring facility”, “When the referring facility cannot no more accept patients due to shortage of beds and unavailability of professionals”, and “Referrals are also made to the lower level health facilities and community based organizations in the best interest of the patient depending on the condition of the patient and the capacity of the lower level health facility /community based organization” (FDRE-MoH 2010).   Added. Using health-center RRL categories such as disease name (or cases), signs and symptoms, conclusion, and referral-out reason, we examined the referral cases and the reasons for the referral decisions.   Added. It accounted for 40.08 %, 39.22 %, and 16.34 % of all 459 referrals, respectively. 

In paragraph 3 under results, the issue of timeliness was not duly covered in the table 2: Analysis of RRL data quality metrics:  Thank you for your comment. We have now separated timelines from completeness. Added Timeliness Row on Table 2.  The RRL column records were found to be up to date and reported as such. Referral feedback columns, on the other hand, were usually missing and were not used for real-time reporting.

Can the results also show information outcome of referrals as a proxy appropriateness of referrals based on the condition for treatment? Thanks for raising these question. The hospital will assess the health center RRL-out information and register as well as update the hospital RRL-in. Examining the RRL column "the final remark and feedback," which was filled by the referring hospital and forwarded to the health center for future referral reference and quality improvement, helped establish the appropriateness of the referral. The referral feedback is crucial in establishing whether or not the referral case is correct. Unfortunately, this feedback and remark was lost in the health center's RRL. We can only verify the appropriateness of referrals based on hospital RRL.

Discussion:

Reviewer 1:

In page 15, line 3, there should be a space between results, and 11/12. Thank you for the comment. We added Space

In page 15 line 7, the % sign was omitted from the 72. Thank you for the comment. Added: %

In page 15 under the discussion, again this does not cover the entirety of the study objectives tough it clearly highlights the bottleneck of referrals as the non-provision of feedbacks from the referral point to the referring facility. We appreciate that the reviewer brought this up. We started our discussion with an overview of data quality evaluation. Then, we describe how the referring healthcare practitioner makes the referral choice (evaluating the "signs and symptoms" and "diseases (or cases") for referral "the conclusion or outcome") and complete the RRL form documentation process. Following that, referral feedback and matching of referral entries between the health center RRL-out and the referring hospital RRL-in are performed. Finally, utilizing existing paper-based referral coordination and communication, we examined the overall history of the referred patients as well as referral feedback.  However, on paragraph two, we added in Section 4 the following for further explanation:  The referring health professional makes the decision and fills out the RRL refer-out reason based on the referral criteria. Prior to making a referral decision, the referring health professional is responsible for filling out the RRL form with the essential information and attaching any relevant documents. The RRL contains information on patient referrals such as "disease name (cases)," "referred to”, "referral reason”, and "collected signs and symptoms" that were used in primary care. RRL categories "signs and symptoms" and "diseases (or cases)" were used to make the referral decision, as well as the conclusion (or outcome). The signs and symptoms were used to assess the risk of diseases (cases), and the referring health professional documented the referral-out justification. Referral decisions are made when the criteria for referral are medically appropriate and in the best interests of the patient or client. Furthermore, the national guidelines are also used as a reference guide to address the most common presenting symptoms and referral criteria, which aids in the prioritization and documentation of the referral condition (FDRE MoH, 2010, FDRE-MoH 2017). 

Reviewer 2: In Page 16 you indicate that the proportion of “matching”. The sentence seems incomplete. Is it matching entries on referral?

Thanks for the comment and update as …. the proportion of matching entries on referral between health center and referred hospital RRL was 13.9% only)

In the second paragraph in the discussion section you mention that the reasons for referral do not match the guidelines. In my view – the intention is not to exactly match the guidelines word for word. Whereas the reasons for referral in the RRL book indicate reasons that you mention such as “better patient management” they convey the nuance of the needs of the healthcare worker and should hence not be seen as a radical departure. The guidelines should be seen as guidelines and not an absolute way of filling in a referral form. I suggest that you discuss the evidence on the practical use of the form as a way of describing the departure from the guidelines.

We appreciate that the reviewer asked the intention is not to exactly match the guidelines word for word. Despite the fact that the guidelines are not an absolute means of filling out a referral form, it is anticipated that the RRL documentation is based on the national guidelines. This minimizes the variances between health-center referral-out reasoning and referring hospital referral-in reasoning. We added an explanation of the referral practice including the reason and criteria for the referral. The updated version is listed below. 

We added: 

The referring health professional makes the decision and fills out the RRL refer-out reason based on the referral criteria. Prior to making a referral decision, the referring health professional is responsible for filling out the RRL form with the essential information and attaching any relevant documents. The RRL contains information on patient referrals such as "disease name (cases)," "referred to”, "referral reason”, and "collected signs and symptoms" that were used in primary care. RRL categories "signs and symptoms" and "diseases (or cases)" were used to make the referral decision, as well as the conclusion (or outcome). The signs and symptoms were used to assess the risk of diseases (cases), and the referring health professional documented the referral-out justification. Referral decisions are made when the criteria for referral are medically appropriate and in the best interests of the patient or client. Furthermore, the national guidelines are also used as a reference guide to address the most common presenting symptoms and referral criteria, which aids in the prioritization and documentation of the referral condition (FDRE MoH, 2010, FDRE-MoH 2017).

In the fourth paragraph pg. 16 revise spelling for found. The sentence on the limitations also needs to start in its own paragraph.

Thanks for the comment. The spelling “found” updated and the updated looks like. Unfortunately, we did not find a referral made to lower-level health facilities. The limitations now starts in a new paragraph. 

Conclusions:

In page 17 paragraph 2 the conclusion indicates that the study addressed the issues of how the referrals decisions were made, the coordination and communication of the referrals but these were not presented in the results and further discussions.  We appreciate the motivation for these comments. We have clarified and added the clarification in the Results and Discussion to answer how the referral decisions (the reason for the referral) were made .

In paragraph 2 line 12, the recommendation did not target a specific stakeholder. We appreciate the motivation for these comments. We added “in primary care settings”

References:

We appreciate the positive comments from Reviewers. In response to [18-25], we rectified all references and citations. The citation is created automatically using the Harvard reference style in Latex/Overleaf. The citation is now consistent in terms of formatting, capitalization, space, punctuation, usage of colon and semicolon, and so on. For instance, we use the format of "FDRE-MoH 2016”, "Teklu et al. 2020”, and "Bettencourt-Silva, Clark, et al. 2015; Zhang, Padman, and Patel 2015; Bettencourt-Silva, Mannu, and Iglesia 2016; Funkner, Yakovlev, and Kovalchuk 2017; Zhang and Padman 2017” for “single author citation”, “multiple authors citations”, and ”multiple sources in one reference” respectively. Also, organized chronologically from oldest to newest.

In page 9, line 5, amia2011: Authors should consider capitalizing A and provide a space between Amia and 2011 The citation is corrected and updated

In page 9, line 8, Teklu, et al 2020: I suggest the punctuation should come after the et al. Updated as: (FDRE-MoH 2016; Teklu et al. 2020)

In page 9, line 10, Heimly2011: The authors should provide a space between the author’s name and year of publication and a semi-colon after the year of publication. Updated as: (Heimly and Nytrø 2011)

In page 9, line 18, Fehlings & ODonnel: should be stated Fehlings and ODonnel. Updated as: (De Bleser et al. 2006; Vargus-Adams Glader Fehlings and ODonnel 2017). Also, organized chronologically from oldest to newest.

In page 9, line 20, (Chu, 2001a, Chu, 2001b, DiJerome, 1992 and Donald et al., 2016)): consider the arrangement of the year of publications. I suggest it should start from the oldest to the current year of publication. Also, please be consistent with the typing of the et al, it's either you are using a full stop together with a comma or choosing one of the afore-mentioned punctuation signs throughout. Again, consider closing the in-text reference with just a bracket. Updated as: (DiJerome 1992; Chu 2001a; Chu 2001b; Donald et al. 2016). Also, organized chronologically from oldest to newest.

In page 9, line 22, Gonz´alez-Ferrer et al. ,2013: Be consistent with formatting, the et al issue again. Updated as: Gonz ´alez-Ferrer et al. 2013

In page 9, line 24 and 25, (Bettencourt-Silva, Mannu, and Iglesia, 2016; Zhang and Padman, 2017; Bettencourt-Silva, Clark, et al., 2015; Funkner, Yakovlev, and Kovalchuk, 2017; Zhang, Padman, and Patel,2015): Bettencourt-Silva has no year of publication and I suggest the years are arranged from the oldest to the newest. Updated as: (Bettencourt-Silva, Clark, et al. 2015; Zhang, Padman, and Patel 2015; Bettencourt-Silva, Mannu, and Iglesia 2016; Funkner, Yakovlev, and Kovalchuk 2017; Zhang and Padman 2017). Also, organized chronologically from oldest to newest.

In page 11, line 9 (under Study Setting and Design): (Daniel, W. W., and Cross, C. L., 2018; Israel, G. D., 1992): In-text referencing should be stated appropriately. Updated as: (Israel 1992; Daniel and Cross 2018). Also, organized chronologically from oldest to newest.

---

## [Decision Letter · Decision Letter 1]

16 May 2022

PONE-D-21-20872R1Analysis of referral pathways between low resource health settings to improve coordination and evidence-based services for maternal and child health in EthiopiaPLOS ONE

Dear Dr. Tegenaw,

Thank you for submitting your manuscript to PLOS ONE. After careful consideration, we feel that it has merit but does not fully meet PLOS ONE’s publication criteria as it currently stands. Therefore, we invite you to submit a revised version of the manuscript that addresses the points raised during the review process.

We look forward to receiving your revised manuscript.

Kind regards,

Jackline Oluoch-Aridi, Ph.D.

Guest Editor

PLOS ONE

Journal Requirements:

Additional Editor Comments:

The title seems to be still misleading and insinuates a comparison but does not clearly articulate the respective entities that are under comparison. Authors need to revise to provide clarity.

Reviewers' comments:

Reviewer's Responses to Questions

**Comments to the Author**

1. If the authors have adequately addressed your comments raised in a previous round of review and you feel that this manuscript is now acceptable for publication, you may indicate that here to bypass the “Comments to the Author” section, enter your conflict of interest statement in the “Confidential to Editor” section, and submit your "Accept" recommendation.

Reviewer #3: All comments have been addressed

2. Is the manuscript technically sound, and do the data support the conclusions?

Reviewer #3: Yes

3. Has the statistical analysis been performed appropriately and rigorously? 

Reviewer #3: Yes

4. Have the authors made all data underlying the findings in their manuscript fully available?

Reviewer #3: (No Response)

5. Is the manuscript presented in an intelligible fashion and written in standard English?

Reviewer #3: Yes

6. Review Comments to the Author

Reviewer #3: There is a problem with the title. The use of the word' between' suggests that more than one low resource setting was studied, whereas the study was conducted in Ethiopia only.

Please either use 'low resource setting' or 'poor resource setting'

In the conclusion, you suggest use of automated patient management as a solution- is the infrastructure available? is the human resource trained? what would be the 'cost benefit' implication? Whereas you mention that this is inline with the clinical guidelines, context is very important

7. PLOS authors have the option to publish the peer review history of their article (what does this mean?). If published, this will include your full peer review and any attached files.

Reviewer #3: **Yes: **Many M. Nyikuri (PhD)

---

## [Author Response · Author response to Decision Letter 1]

21 Jun 2022

Reviewer #3: There is a problem with the title. The use of the word' between' suggests that more than one low resource setting was studied, whereas the study was conducted in Ethiopia only. 

We appreciate the efforts by you and the reviewers on the manuscript. To better frame the main thesis of our paper, we updated the title to “Analysis of low resource setting referral pathways to improve coordination and evidence-based services for maternal and child health in Ethiopia”

Please either use 'low resource setting' or 'poor resource setting’

Thanks for the reviewers comment and we appreciate your suggestion. Now, we updated the 'poor resource setting' to 'low resource setting’.

In the conclusion, you suggest use of automated patient management as a solution- is the infrastructure available? is the human resource trained? what would be the 'cost benefit' implication? Whereas you mention that this is inline with the clinical guidelines, context is very important. 

We appreciate the motivation for these comments. We updated and provided context. We added “when the necessary infrastructure and human resources are in place”. The whole sentences are as follows:

Therefore, when the necessary infrastructure and human resources are in place, implementing an automated patient management system may help to correct and impose a standardized referral service workflow.

Finally, to provide additional explanation and context, we added:

Furthermore, even before viable infrastructure is implemented and competent human resources are attracted, a minimal infrastructure (a low-cost alternative) and trained personnel can facilitate referral coordination and offer evidence-based services to ensure a buy-in and acceptance by all stakeholders. This transition needs careful planning and openness to adopt best practices in order to reduce resource waste, to ensure training, maintenance, and avoid heavy service expenses.

---

## [Editor Report · Decision Letter 2]

9 Aug 2022

Analysis of low resource setting referral pathways to improve coordination and evidence-based services for maternal and child health in Ethiopia

PONE-D-21-20872R2

Dear Dr. Geletaw Sahle Tegenaw

We’re pleased to inform you that your manuscript has been judged scientifically suitable for publication and will be formally accepted for publication once it meets all outstanding technical requirements.

Kind regards,

Jackline Oluoch-Aridi, Ph.D.

Guest Editor

PLOS ONE

Additional Editor Comments (optional):

To the authors,

The article has undergone significant improvements with the authors acknowledging the feedback of our reviewers.
---

## [Editor Report · Acceptance letter]

17 Aug 2022

PONE-D-21-20872R2 

Analysis of low resource setting referral pathways to improve coordination and evidence-based services for maternal and child health in Ethiopia 

Dear Dr. Tegenaw:

I'm pleased to inform you that your manuscript has been deemed suitable for publication in PLOS ONE. Congratulations! Your manuscript is now with our production department. 

Kind regards, 

on behalf of

Dr. Jackline Oluoch-Aridi 

Guest Editor

PLOS ONE